# Macro- and Microstructure of In Situ Composites Prepared by Friction Stir Processing of AA5056 Admixed with Copper Powders

**DOI:** 10.3390/ma16031070

**Published:** 2023-01-26

**Authors:** Valery Rubtsov, Andrey Chumaevskii, Anastasija Gusarova, Evgeny Knyazhev, Denis Gurianov, Anna Zykova, Tatiana Kalashnikova, Andrey Cheremnov, Nikolai Savchenko, Andrey Vorontsov, Veronika Utyaganova, Evgeny Kolubaev, Sergei Tarasov

**Affiliations:** Institute of Strength Physics and Materials Science, Siberian Branch of Russian Academy of Sciences, 634055 Tomsk, Russia

**Keywords:** friction stir processing, dissimilar, intermetallic compounds, metal matrix composites, copper powder, mechanical properties, structural phase state, solid solution

## Abstract

This paper is devoted to using multi-pass friction stir processing (FSP) for admixing 1.5 to 30 vol.% copper powders into an AA5056 matrix for the in situ fabrication of a composite alloy reinforced by Al-Cu intermetallic compounds (IMC). Macrostructurally inhomogeneous stir zones have been obtained after the first FSP passes, the homogeneity of which was improved with the following FSP passes. As a result of stirring the plasticized AA5056, the initial copper particle agglomerates were compacted into large copper particles, which were then simultaneously saturated by aluminum. Microstructural investigations showed that various phases such as α-Al(Cu), α-Cu(Al) solid solutions, Cu_3_Al and CuAl IMCs, as well as both S and S’-Al_2_CuMg precipitates have been detected in the AA5056/Cu stir zone, depending upon the concentration of copper and the number of FSP passes. The number of IMCs increased with the number of FSP passes, enhancing microhardness by 50–55%. The effect of multipass FSP on tensile strength, yield stress and strain-to-fracture was analyzed.

## 1. Introduction

Friction stir processing is a promising and growing technology for fabricating metallic matrix composites (MMC) using different metal combinations [1,2]. Using this technology, it becomes possible to manufacture and modify various MMCs such as, for example, Al/Al_3_Ti by FSP on a preliminarily pre-compacted Ti-Al powder mixture when Al_3_Ti particles appear from the Ti-Al exothermic reaction [3]. The FSP on the as-cast Al-Mg/Ti-B MMC was carried out to enhance its strength characteristics by grain refining, which also resulted in an enhanced corrosion resistance [4]. The multipass FSP on a copper plate with a groove filled with SiC particles was used to obtain Cu/SiC composites characterized by higher strength and electric resistivity, as well as friction reduction [5]. An analogous but single-pass FSP method was used for producing Cu/Al_2_O_3_, Cu/B_4_C, and Cu/TiC wear-resistant composites [6].

Friction stir processing and intermixing SiC particles loaded into holes drilled in a Ti-6Al-4V alloy plate resulted in a nearly twofold increase in microhardness and wear resistance [7]. The multipass FSP intermixing of copper powder into Ti-6Al-4V allowed fabricating an in-situ composite reinforced both by Ti_2_Cu particles and spinodal decomposition, that demonstrated reduced wear and friction [8]. The magnesium matrix composite Al-Ti-TiC-CNTs/AZ31 was prepared using the high-speed (3000 RPM, 1000 mm/min) FSP on a coating made of mechanically alloyed and laser pre-sintered Al-Ti-TiC-CNTs powder composition, demonstrated the presence of the intermetallic particles Al_12_Mg_17_ and TiC, which enhanced both hardness and corrosion resistance [9].

As-casted Mg/2Si composites were subjected to multipass FSP, resulting in the formation of both primary and eutectic Mg_2_Si particles, improved stability of the tribological layer and reduced mass loss in sliding [10]. The FSP has also been used for intermixing of an as-cast high-entropy AlCoCrFeNi_2.1_ alloy with SiC particles [11]. The role of copper additions in the transformation-induced plasticity (TRIP) high entropy alloys was related to the increase in the amount of ε-hcp phase, thus enhancing the TRIP effect [12].

Friction stir processing of low-carbon steels was accompanied by intense interaction with the WC-Ni-base FSP tool and transfer of the wear debris into the stir zone (SZ) [13].

Many other aspects of the effect of FSP on steel, including tensile behavior, fracture toughness, corrosion and wear behavior, may be obtained elsewhere [14].

Surface hardening of the aluminum alloy was carried out by FSP admixing of Mo powder to an Al-Mg-Sc alloy [15] or by in situ formation of intermetallic Al_3_N nanoparticles in AA7075 [16], as well admixing zirconia to AlZnMgCu1.5 [17]. The addition of either ZrO_2_ particles [18] in FSP served to reduce the composite wear rate by 62% compared to that of the base metal. Adding SiC-graphite composition to AA7075 improved the wear resistance by 78% [19]. A standalone trend may be the use of FSP for hybrid hardening of metallic materials, with the aim of obtaining their improved functionality, as shown by the example of FSP admixing of carbon nanotubes into an Al-Zr matrix [20].

There are a number of methods used for the surface modification of metallic materials with the use of FSP. The simplest one is to use FSP without admixing/synthesizing any reinforcing or hard particles/precipitates into a base material except for strain-induced dissolving and precipitation of particles inherent with the alloy; for example, Al_3_Zr in an Al-Mg-Sc-Zr alloy [21]. On plain ductile metals capable of only grain boundary strengthening, this approach makes it possible to obtain the fine-grained structures with improved strength but impaired ductility [1]. This approach may also be applied to coarse-grained materials as those obtained via additive manufacturing [22].

A combination of hardening mechanisms such as grain boundary (the Hall–Petch law) and martensitic transformation may be achieved using the FSP on carbon steels to obtain hard and wear-resistant quenched structures [23]. Hardening the aluminum or titanium alloys requires another FSP approach that would imply admixing the components capable of the in situ formation of intermetallic compounds with the base metal. For example, this approach was used in an overlap FSP on copper/aluminum alloy or copper alloy/aluminum alloy sheets and had a dramatic structural modification effect on the base materials [24]. The disadvantage of FSP on dissimilar metal sheets is the inhomogeneous structure of the stir zone and the formation of large and brittle intermetallic layers.

The third approach involves FSP admixing the powder particles into the base metal in order to form composite structures [25,26]. To enable such a process, the powder must be first delivered to the base metal workpiece using any suitable technique such as, for example, loading the powder either into cut grooves [25] or into a series of drilled holes [25,27]. The variable width groove technique may also be applied for obtaining gradient and shaped structures [28]. Meantime, the above-described drilled hole technique is simple and widely used for preparing composites from a number of dissimilar alloys and metals by means of FSP [29]. The moving FSP tool shoulder caps the hole before the FSP tool pin has even started the intermixing, while in the case of using the milled grooves, they are closed beforehand to avoid the premature squeezing out and spilling of the powder.

Despite that, the formation of the FSP composite structure is not fully understood in terms of powder intermixing and interaction with the plasticized metal flow, especially if large powder concentrations were used. What is known so far is that the powder is carried and then distributed by means of very intricate metal flow trajectories [15,30].

The plasticized metal transfer in FSP is possible mainly due to the adhesion of the plasticized metal to the FSP tool surface [21] and the extrusion of this metal from the leading zone into the rear one behind the tool [31]. This plasticization of the base metal in FSP is achieved by frictional heating and severe plastic deformation of the base metal with the generation of nanosized grains capable of superplasticity. In addition, the strain-induced dissolution of various intermetallic particles and inclusions occurs during FSP and serves to create a saturated solid solution in the stir zone, which then experiences precipitation and recrystallization [32]. In addition, various structural and phase transformations may occur if allowed by the phase diagrams of the metals processed.

The plasticized metal flow patterns in FSP were studied by admixing immiscible metal and then examining its distribution using the X-ray transmission methods, including computer tomography [33,34,35]. The intricate flow patterns were revealed as determined by different deformation, temperature and stirring conditions established on the advancing (AS) and retreating (RS) sides of the stir zone [36]. Other important factors may be plastic strain and flow rate gradients formed along the tool radius direction in the base material [33]. Additionally, there are such factors as upward and downward metal flows [37,38,39].

The single-pass FSP does not provide the stirring efficiency acceptable for the homogeneous distribution of components [40], therefore there is need for multi-pass FSP [41]. Another factor important for homogeneous powder distribution in the stir zone is the amount of powder intermixed [15].

There is a lack of studies devoted to dynamic and pervasive descriptions of FSP plasticized metal transfer with dissimilar material particles, including descriptions of possible in situ structural and phase transformations, dissolution and precipitation of new phases, as well as interactions with the FSP tool material. This task, however, is closely related to parametrical studies of the process, i.e., establishing a relationship between the parameters of FSP, such as tool rotation frequency, plunging force, traverse speed, additional energy input, cooling conditions, etc., and the structural response of the stir zone material. The primary goal here could be experimental modeling of composite formation in FSP, such as by admixing copper powder into an aluminum alloy matrix. This option may be preferable for a number of reasons. First of all, there is significant difference between their physical characteristics that allows for a good contrast in X-ray radioscopic, back scattering electron (BSE), scanning electron microscopy (SEM) and TEM images. Secondly, the aluminum/copper exothermic interaction in powder systems is well-known, as well as corresponding phase diagram. Previous studies showed that admixing copper particles into an aluminum alloy matrix resulted in exothermic diffusion-reaction and the formation of Cu-Al intermetallic compounds, for example Cu_2_Al [42,43,44].

Although many publications relate to the in-situ obtaining the MMCs using the FSP, there is a lack of publications devoted to the efficiency of stirring and phase formation processes as dependent on the concentration of the added material and number of FSP passes.

The objective of this research is to study the efficiency of copper intermixing with the AA556 stir zone metal vs. copper concentration and number of FSP passes, as well as to characterize the effect of copper addition on the mechanical characteristics of the stir zone metal.

## 2. Materials and Methods

The FSP was carried out using the facilities and equipment provided by the Institute of Strength Physics and Materials Science SB RAS, Tomsk, Russia) (Figure 1a). The workpieces in the form of plates were cut off a 5 mm-thick AA5056 rolled sheet. The 3 mm in depth holes were then drilled in them, as shown in Figure 1b. These holes were filled with densified copper powder so that the total powder contents were 1.5, 3.0, 6.0, 15 and 30 vol.%, depending on the number and diameter of the holes drilled in each of the plates (Figure 1b). The powder contained 10.5 ± 0.5 μm particles of 99.5% pure copper Cu.

FSP was performed using a 3 mm high cone, a threaded pin with top and bottom diameters of ∅6 and ∅8 mm, respectively, and a ∅16 mm shoulder, as shown in Table 1. The values of the parameters such as plunging force P_z_, tool angular velocity ω_z_, and tool traverse speed V_x_ were borrowed from friction stir welding (FSW) on an AA5056. Then, the P_z_ values were tailored to exclude the excess penetration of the FSP tool into the workpiece because of extra heating and plasticization. In particular, the plunging force 13.5 kN provided steady FSP, both in FSW on AA5056 and in FSP admixing 1.5–6.0 vol.% of copper powders. However, excess plasticization was noticed in the initial FSP passes on stirring samples with 15 and 30 vol.% of Cu. Therefore, the first passes were carried out at P_z_ values reduced to 12.5 kN. However, further FSP passes required increasing the P_z_ values. The mechanical response of the stirred material to FSP was determined by measuring torque M_z_ and reaction force to the traverse movement P_x_ (Figure 1a). Table 1 also shows the mean mechanical characteristics of the stir zone metal such as ultimate tensile strength (UTS), yield stress (YS) and strain-to-fracture (STF).

The samples for mechanical and metallographic investigations were cut off the workpieces using an EDM machine DK7750 (Suzhou Simos CNC Technology Co., Ltd. Suzhou, China) as shown in Figure 1c. Tensile tests were performed on the dog-bone samples with their tensile axis oriented along the FSP track axis and using a test machine UTS110M (Testsystems, Ivanovo, Russia). The microhardness profiles were obtained using an Affri DM8 microhardness tester (Affri, Italy) by indenting across the stir zone and along the RS-to-AS midline (Figure 1d, line AB).

Microstructural examination of the samples was carried out using an optical metallographic microscope Altami MET-1C (Altami, St. Petersburg, Russia), confocal microscope OLYMPUS LEXT (Olympus NDT, Inc., Waltham, MA, USA)), an SEM instrument Zeiss LEO EVO 50 (ZEISS, Oberkochen, Germany), TEM instrument JEOL-2100F (JEOL Ltd., Tokyo, Japan) and an X-ray computer tomography YXLON Cheetah EVO (YXLON International GmbH, Hamburg, Germany). Samples for mechanical and metallographic optical as well as scanning electron microscopic investigations were cut off the workpieces using an EDM machine DK7750 (Suzhou Simos CNC Technology Co., Ltd. Suzhou, China), as shown in Figure 1c. Thin foils for TEM studies were prepared using an EM-09100IS (JEOL Ltd., Japan) sample preparation system.

The phase composition was studied by X-ray diffraction analysis using an XRD instrument DRON 7, CoKα (Bourevestnik, JSC, Saint-Petersburg, Russia).

## 3. Results

The time dependencies of both torque and reaction force were recorded to characterize the power characteristics of FSP. The torque M_z_ values mainly determine the frictional heat input while the reaction force P_x_ accounts for the plasticized metal resistance to FSP and, thus, may characterize its effective viscosity and strain rate as functions of temperature and grain size. The FSP on pure AA5056 is characterized by the reaction force vs. time dependencies with only slight oscillations. The highest mean P_x_ values were obtained during the first FSP pass and then reduced with the pass number (Figure 2a). The same is true for FSP on the 1.5 vol.%Cu and 3.0 vol.% Cu samples but there is an even larger difference between the first and further passes (Figure 2b and Figure 3c). Additionally, the dependencies allow observing the larger P_x_ oscillations in comparison to those in Figure 2a,c. The difference in P_x_ between the first and the following passes becomes less noticeable in FSP on samples with higher concentrations of Cu (Figure 2d–f). The oscillations become even larger and finally impose on each other, thus forming a wide scatter region, i.e., the traverse movement of the tool allows it running periodically against copper agglomerates, i.e periodically overcoming some barriers that may be related to insufficient plasticization of the copper-rich regions.

The first pass oscillations may be directly related to interacting with copper-filled holes so that the more holes the more reaction force peaks (Figure 2g). At the same time, the mean values of reaction force during the first pass are reduced, especially at 15 and 30 vol.% samples. The same tendency exists also for the fourth pass dependencies, where, despite the increase in oscillation peaks, the mean reaction force values decrease with the increase in copper content (Figure 2h).

It was shown earlier [45] that the reaction force is decreased when the FSP tool meets the earlier metal after FSP with the micron-sized recrystallized grains, and no extra energy is spent for deformational grain refining.

The above-discussed dependencies do not afford unambiguous interpretation. For example, FSP of AA5056 and samples with 1.5, 3.0, and 30 vol.% of Cu tended to reduce the reaction force as the pass number increased (Figure 3a). For the 6.0 and 15.0 vol.% Cu samples, this tendency was not so obvious. The reaction force oscillation peaks, however, demonstrate some reduction vs. the pass number for all the samples except one with 30.0 vol.% Cu (Figure 3b). This tendency for reducing the material resistance with the pass number is retained in case of measuring the torque, and, on the contrary, there is no clear dependence on the copper concentration (Figure 3c).

The above-described dependencies describe the dynamics of stirring that should be have their effect on macro- and microstructures of the stir zone formed after each FSP pass.

The single FSP pass on the workpiece with 1.5 vol.% Cu load resulted in the formation of minor inhomogeneity in the Cu powder distribution stir zone; however, this disappeared after the 2-pass FSP (Figure 4a). The advancing side of the stir zone looks more dark-etched in comparison to the retreating side. Such a difference may be related to better plasticization of both components and dissolution of copper on that side compared to those on the advancing side, where copper particles are still agglomerated and cold.

An analogous macrostructural pattern can be observed in samples with 3.0 vol.% Cu where the advancing side looks more heavily etched after the first FSP pass (Figure 4b). Large compacted copper areas can be observed after pass #3; however, they disappear after four FSP passes. Similar macrostructures were observed in samples containing 6 vol.% Cu (Figure 4c), except for the fact that large compacted copper chunks formed just after the first FSP pass. Increasing the copper content up to 15 vol.% resulted in the formation of a large compacted copper area on the advancing side after the first pass (Figure 4d). Further FSP passes served for breaking it into the smaller ones so that none of them have been observed after pass #4. The 30 vol.% Cu sample is characterized by greater inhomogeneity, in which even the 4-pass FSP is not effective for full homogenization of the compacted copper.

It can be observed that the compacted copper particles form streaks oriented with respect to the plasticized metal flow (Figure 5), and one may assume that the stir zone microstructures are formed by these plasticized AA5056 metal flows, which carry and dynamically compact the copper particle agglomerates into sintered copper chunks with intrinsic diffusion and diffusion-reaction between aluminum and copper.

Therefore, these compacted copper-rich particles can be represented by an α-Cu(Al) aluminum solution in copper and also contain grey areas that may be Al/Cu intermetallic compounds (IMC). For small copper concentrations, a stirring-compaction-diffusion process is the most effective for providing a rather homogeneous distribution of smaller copper-rich compacted particles in the stir zone after 4-pass FSP (Figure 5a). Although the size of these compacted particles decreases with the increase in the number of passes, there may still be a number of large ones, such as those shown in Figure 5a. In FSP, a similar situation can be found when admixing 3.0 vol.% Cu to the AA5056 (Figure 5b), but in this case there are more compacted larger copper-rich particles. After 4-pass FSP, almost all these particles acquired a grey color, i.e., they were transformed into IMCs.

In fact, increasing the copper concentration in the AA5056 matrix up to 6 (Figure 5c) and 15 (Figure 5d) vol.% Cu resulted in an increase in the number and size of the compacted copper/aluminum particles, so that even four passes may not be enough to obtain more or less homogeneous distribution throughout the stir zone. The most inhomogeneous microstructure was formed in the 30 vol.% Cu sample (Figure 5e), obviously due to poor heating and stirring the two-component mixture. However, the metal flow remains the main factor for copper/aluminum particle structure formation by mechanical compaction, dynamic sintering and diffusion.

All the above-discussed results allow suggesting that the single-pass FSP is not enough to initiate the most intensive structural and phase changes. The X-ray diffraction (XRD) pattern obtained after both the first and forth FSP passes on the 1.5 vol.% Cu sample only showed the α-Al(Cu) peaks (Figure 6a,b). The 1-pass FSP on the 3 to 30 vol.% Cu samples allowed the formation of α-Cu (Al) peaks, in addition to the α-Al (Cu) ones (Figure 6a). So far, no IMC peaks have been detected in these diffractograms. This suggests that the exothermic reaction-diffusion between Al and Cu was not intensive enough, despite the fact that the reaction force values corresponding to the first FSP pass in Figure 3a are less than that of the 1.5 vol.% Cu. Regarding 4-pass FSP, all the diffractograms obtained from 3 to 30 vol.% Cu samples demonstrated the presence of both α-Al(Cu) and α-Cu(Al), as well as Al_2_CuMg and AlCu_3_ IMCs, the content of which increased with that of copper powder (Figure 6b).

Mechanical characteristics such as ultimate tensile stress (UTS), yield stress (YS) and strain-to-fracture (STF) (Figure 7a–c) are presumably enhanced with the FSP pass number and the stir zone homogeneity. The UTS and YS become higher with the pass number for all the samples, except for those that are macrostructurally inhomogeneous even after 4 passes 30 vol.% Cu sample (Figure 7a,b). Nevertheless, the UTS of AA5056 after undergoing FSP stays absolutely maximum and increases with the pass number. The UTS of the 4-pass FSP 6.0 vol.% Cu sample is the closest to that of AA5056 (Figure 7a).

The YS value is maximum for the 4-pass FSP 15 vol.% and 6 vol.% Cu samples, i.e. higher than that of the AA5056 (Figure 7b) after the FSP, the corresponding STF values are maximum for all the pass numbers (Figure 7c). The STF values of composite samples increase with the pass number but always stay below those of FSPed AA5056. The maximum STF value among all the composites was achieved on the 1.5 vol.% Cu sample after 4 passes. The most noticeable STF decrease occurred in samples with copper content from 3 to 30 vol.%.

The microhardness profiles were obtained as shown in Figure 1d, depending on the copper content and number of passes (Figure 7d–i). The FSP on AA5056 did not result in any microhardness level enhancement, irrespective of the pass number (Figure 7d). The microhardness number profile is almost monotonous within the 0.8–0.9 GPa interval.

Admixing 1.5 vol.% Cu resulted in average microhardness enhancement by 20% (Figure 7e) in comparison to that of the AA5056 after FSP. The microhardness number scatter is within the 0.75–1.05 GPa interval without any high peaks, the appearance of which was noticed in samples containing 3–30 vol.% Cu (Figure 7d–i) with numerous IMCs (Figure 5b–e). The mean microhardness level is also increased up to 50–55% with the copper content and taking into account the degree of structural inhomogeneity.

The above-discussed macroscale inhomogeneity and copper particle distribution has been studied using the X-ray computer tomography (XCT) and metallographic views prepared in different sections. The XCT images of samples with 15 (Figure 8a) and 30 (Figure 8b) vol.% Cu show that after the first pass, copper powder was concentrated on the retreating sides mainly in the form of large agglomerates. When using the two rows of drilled holes for admixing 30 vol.% Cu (Figure 8b), the copper distribution pattern is changed so that some copper is left on the advancing side too.

Further FSP passes resulted in a more homogeneous distribution of copper by the FSP track width in the stir zone, while the retreating side still contained more copper (Figure 8a,b). Such a finding may be explained by the less efficient adhesion transfer due to the admixing of copper particles to the plasticized base metal.

Higher magnification XCT images obtained from the 6.0 vol.% Cu and 15 vol.% Cu samples demonstrate that some periodical transfer pattern is formed in the 6.0 vol.% Cu sample after the first FSP pass (Figure 9a), again with a concentration of copper on the retreating side. In the case of the 15 vol.% Cu sample, the concentration of copper on the retreating side is an even more prominent transfer (Figure 9b). An acceptable homogeneous distribution of copper was achieved after the 4-pass FSP in the 6.0 vol.% Cu sample with only minor copper agglomerations on the retreating side. However, four passes were not enough for the homogenization of the 15 vol.% Cu sample, where coarse particles were present in the stir zone together with the copper agglomerates on the retreating side.

Taking into account the XCT images obtained from the 6.0 vol.% Cu sample, it could be suggested that the copper particle dispersion in the AA5056 stir zone occurs synchronously with the plasticized metal transfer, i.e., with highly efficient stirring provided by the combination of adhesion transfer and extrusion (Figure 10).

Horizontal sections prepared on the FSP tracks of the 1.5 vol.% Cu sample (Figure 11a) revealed narrow compacted copper streaks elongated in the transfer direction, which then dissolve in further FSP, thus leaving some isolated small particles. A higher concentration of copper in the 6.0 vol.% Cu sample (Figure 11b) resulted in an increase in both the number and size of the compacted copper streaks. The size of the streaks is also dependent on the stirring intensity by the plasticized metal flows. For example, the metal flows may collide with each other and form eddy zones where deformation and transfer rates would be minimal and where large copper agglomerates would settle down. It is reasonable that agglomerates form in the vicinity of the thermomechanically affected zones where stirring is weak and grain growth may dominate over grain segmenting by deformation [45]. Another example of such a stagnant zone may be a place where metal flows driven by the tool’s pin and shoulder collide. It is exactly where the wormhole defects appear when using the FSP parameters beyond their optimal range. An example of such a stagnant zone is represented by images in Figure 11c, where large copper particles do not disappear even after the fourth FSP pass.

Apart from pure mechanical breaking of the large compacted particles, there may be plastic deformation, as well as simultaneous diffusion of copper into aluminum. The thermomechanical conditions of stirring such as plastic deformation rate and temperature also have an effect on the diffusion of copper into aluminum, as well as on IMC formation.

The TEM images make it possible to observe the detailed structure of Al-Cu intermetallic particles formed in the stir zones of samples, particularly those with 1.5 vol.% Cu containing dynamically recrystallized α-Al(Cu) grains with sizes in the range of 0.67–4.21 μm and a mean size of 1.86 ± 0.6 μm (Figure 12a). The dark-field images in Figure 12c,d obtained using α-Al_(3–1-1)_ reflection in the SAED pattern (Figure 12b) show coarse α-Al(Cu) grain with dislocations (Figure 12c). Apart from the α-Al grains, there are reflections identified as those of β-Al_3_Mg_2_ IMC (Figure 12b,d) and S-phase (Al_2_CuMg) (Figure 12e–g).

It is obvious that the images in Figure 12 were obtained from an aluminum-rich area where the local concentration of copper was not high enough to form an Al-Cu IMC. An example of a copper-rich area is represented in Figure 13a in the form of an Al-Cu IMC thin streak surrounded by α-Al(Cu) grains. The EDS elemental maps show a high concentration of copper in this streak, which was formed by compaction and reaction-diffusion in the course of plasticized metal flow. In the region of copper particles, the Guinier–Preston–Bagaryatsky (GPB) zones can be formed (Figure 13a).

The IMC formations in the sample with 6 vol.% Cu look rather as IMC agglomerates composed of smaller IMC grains (Figure 13b). The same is even more obvious with the TEM images of IMCs in the 15 vol.% Cu sample (Figure 13c). At the same time, the EDS maps show some thin copper-rich streaks in them that may represent the unreacted copper core (Figure 13c). The concentration of copper is maximum in the IMCs, which are surrounded by α-Al(Cu) solid solution grains with some smaller IMC precipitates (Figure 14a–c) formed as a result of deformation and breakage of large compacted copper streaks by the plasticized metal flows.

## 4. Discussion

The multipass FSP method was used for intermixing copper powder with the AA5056 stir zone metal with an aim to obtain Al-Cu intermetallic compound particles homogeneously distributed in the fine-grained stir zone matrix. The parameters of FSP such as axial plunge force, tool rotation rate and traverse speed were used almost the same as those previously found for AA5056 friction stir welding but corrected for the possible excess heating and enhanced effective viscosity of the stirred metal. Simultaneously with FSP, the plasticized metal response to stirring was recorded in the form of FSP tool torque and reaction force dependencies on the time and number of passes. The reaction force slightly decreased vs. the number of passes for all the copper concentrations in accordance with the earlier observed and commonly known results [45]. The same behavior was observed for the tool torque values.

These reaction force dependencies on time were characterized by force fluctuations, the amplitudes of which grew with the copper concentration in the AA5056 stir zone and could be related to the inhomogeneous stirring conditions following the computer tomography and metallographic images successively obtained after each FSP pass. The fluctuation may appear when the copper–rich agglomerates interfere with the plasticized metal flow. Meanwhile, copper exothermically reacts with the plasticized aluminum and forms IMCs, thus releasing some additional heat and additionally plasticizing the neighboring stir zone regions.

The metallographic studies were carried out using the cross section metallographic views of the stir zone metal, showing how effective the multipass FSP was for copper dispersion in the stir zone. The results of this investigation showed that the structural evolution of the AA5056 stir zone metal in FSP, admixed with the 1.5–30 vol.% of Cu powder, is determined by the plasticized metal transfer conditions and flow patterns, which in turn, depend on the rheological characteristics of the plasticized AA5056 + Cu metal, concentration of copper, number of FSP passes, etc. (Figure 15 and Figure 16). In the course of the first pass, the FSP tool actuates the previously plasticized AA5056 metal to flow around the tool, as shown in Figure 15, pos. 1. When FSP tool is approached to the copper-filled hole, the copper experiences pressure and heating, resulting in its preliminary compaction. The next stage involves direct friction contact between the tool and pre-compacted copper, during which layer-by-layer friction causes compacting, friction-induced flow and drawing to form the copper streaks in accordance with the metal flow around the tool (Figure 15, pos.2). The traverse movement of the tool provides the layer-by-layer transfer of copper to the retreating side (Figure 15, pos.3) and then to the zone behind the tool (Figure 15, pos.4).

Adhesion of the plasticized metal to the tool surface is an important factor of the metal transfer in FSP, but since the adhesion of copper to the tool is weaker compared to that of more plasticized AAA5056, large friction compacted copper particles settle on the retreating side and only partially transfer to the zone behind the tool (Figure 15, pos.5). Interaction of the FSP tool with the copper-filled holes would create a periodic pattern, as shown in Figure 15, pos.6. This situation changes when copper is derived from the two rows of the copper-filled holes, where copper is more intensively stirred in the row close to the advancing side. However, despite this, the above-discussed considerations regarding the friction compaction of powder still make sense.

The second FSP pass starts from re-plasticizing and re-stirring the centerline SZ regions (Figure 15, pos.7), which contains less copper fragments compared to those on the retreating side. The next stage involves the friction-induced plastic deformation of a large compacted copper chunk and its re-plasticization and transfer closer to the retreating side (Figure 15, pos.8) and even further to the zone behind the tool (Figure 15, pos.9), so that the amount of copper on the retreating side is reduced (Figure 15, pos.10).

The copper transferred into the zone behind the tool becomes more homogeneously distributed and the more FSP passes, the more homogeneous copper distribution is in the stir zone (Figure 15, pos.11). The friction between the tool and copper chunks results in their deformation and fragmentation, especially if these chunks are embrittled by Al-Cu IMCs during previous passes. These small fragments can easily be transferred by the metal flows, while the large ones are still retained on the retreating side (Figure 15, pos.12).

The fourth FSP pass results in further fragmentation of large IMC chunks (Figure 15, pos.13) and the dissolution and precipitation of fine IMC particles, with further homogenization of the metal (Figure 15, pos.14). However, the final structures of the stir zones may differ from each other depending on the amount of copper admixed. Therefore, admixing small quantities results in the formation of small copper-enriched areas located at the periphery of the stir zone, whereas by admixing large copper quantities, the retreating side is enriched by large copper streaks oriented with respect to plasticized metal flow. Samples with 30 vol.% Cu contain, in addition to the copper particles, more dissolved copper on the retreating side of the stir zone.

As shown above, structural evolution during FSP on AA5056 + Cu systems allows revealing its relationship with the force parameters measured on the tool, so that both reaction force and torque reduce with the pass number owing to improved grain structure refining and homogenizing the phase composition. However, there is no direct dependence of these force parameters on the copper content in the composite stir zone. The only finding was that the reaction force reduced with a simultaneous increase in the tool oscillation amplitude when copper powder content increased.

The reaction force reduction may be related to the enhanced plasticization of metal, which occurred due to the extra heat from an exothermic reaction-diffusion between Al and Cu with the ensuing formation of IMCs. The amplitude of the tool oscillation may increase due to instable adhesion between the tool and the components, such as plasticized aluminum alloy and less plasticized copper powder or IMCs. Additionally, the inhomogeneity of the stir zone would enhance the instability of plasticized metal flow.

The above-discussed results make it possible to describe the microstructural evolution of stirring AA5056 with copper powder. The first stage relates to forming copper powder agglomerates, which compact at the second stage under temperature and pressure from the approaching FSP tool, and then by friction between the tool and the pre-compacted copper agglomerate (Figure 16a, pos. I–III). Simultaneously, the compacted particle is deformed in the plasticized metal flow hydrostatically and by rotation under the condition of flow velocity gradient (Figure 16a, pos. IV–V).

In the course of deformation, the large copper particle accumulates defects and thus may experience fragmentation (Figure 16a, pos. VI–VII). The fragments may then be carried away by the metal flows and experience further fragmentation, especially if saturated with aluminum and formed some brittle IMCs (Figure 16a, pos. VIII–IX). These fragments may be found as tracks of small particles located between the metal flow rings around the tool.

The second interesting process is diffusion between Cu and Al with the following formation of solid solution and Al-Cu IMCs (Figure 16b). The intensity of reaction-diffusion in metal flows may be enhanced, so that the IMCs may form from the very beginning of the FSP process despite their presence, is not yet detected by XRD. It was shown earlier that the Al-Cu IMCs may form in the overlap of FSP via a liquid phase, as evidenced by the presence of low-melting eutectics [24]. However, no evidence for the presence of the liquid phase was found in this experiment with the use of copper in the form of powder instead of overlapping copper and aluminum alloy sheets. Therefore, a solid-state reaction-diffusion is proposed here with the formation of IMCs (Figure 16b, pos.I-V). in addition, a solid solution of copper in aluminum is formed where S particles nucleate in accordance with the Guinier–Preston zone hardening mechanism in Al-Cu alloys [46,47] and then grows into large IMCs. The third process can be described as combining large IMC copper-core particle fragmentation and admixing the fragments into the stir zone metal (Figure 16c).

Although the focus of this study was on obtaining the insight into FSP intermixing of copper particles in the aluminum alloy, the mechanical properties of the in situ obtained composites have been determined and compared to those of both as-received and AA5056 after FSP alloys. It was shown that the UTS values obtained on the 4-pass FSP AA5056 + 1.5% Cu, AA5056 + 3% Cu and AA5056 + 6% Cu are 351, 370 and 405 MPa, respectively, i.e., higher than 351 MPa of the as-received AA5056, but, on average, lower than– 395 MPa of the 4-pass AA5056. However, the 4-pass AA5056 + Cu samples after FSP still have some structural and compositional inhomogeneity, as well as microscopic defects. For even higher concentrations of Cu, there is UTS degradation related to insufficient intermixing and structural inhomogeneity of high copper concentrations. It follows from the data in Table 1 that the more Cu concentration, the more passes are needed to achieve acceptable structural and compositional homogeneity. Although the acceptable strength was achieved after 4-pass FSP on 1.5, 3 and 6 vol. % of copper/AA5056 samples, their characteristics can be improved using more FSP passes. The same may be true for samples with 15 and 30 vol.% Cu.

The practical application of Cu-Al materials may be electric copper-to-aluminum connections, which are widely used and whose electric resistance and mechanical strength are greatly dependent on the size of Al-Cu IMCs formed during fusion welding [44]. The use of multipass FSP enables these copper-to-aluminum connectors to be made at considerably lower temperatures and, thus, makes it possible to obtain finer homogeneously distributed IMC particles.

## 5. Conclusion

For FSP, the addition of different amounts of copper powder to the AA5056 has been carried out to study the efficiency of copper powder intermixing and IMC formation depending on the copper concentration and number of FSP passes. The efficiency of AA5056 + Cu plasticization was evaluated by measuring the plasticized metal reaction force, which decreased with the number of passes. The efficiency of copper powder dispersion by the plasticized metal flows in the stir zone was examined using metallographic and X-ray computer tomography views depending on the number of passes and copper concentrations.

Mechanical stirring of the plasticized metal and the interaction between the FSP tool and copper powders make it possible to obtain compacted copper metal formations, even after the first FSP pass, while further passes resulted in the breaking of these formations, reaction diffusion with the aluminum matrix and the formation of Al-Cu intermetallic compounds in the form of homogeneously distributed isolated fine particles, as well as large polycrystalline agglomerates located closer to the shoulder-driven metal zone.

The acceptable homogeneity of the stir zone was achieved after 4 FSP passes for all samples, except for the sample with 30 vol.% Cu. Samples containing 6 vol.% Cu and FSP by 4 passes showed an ultimate tensile stress almost equal to that of AA5056 after FSP.

## Figures and Tables

**Figure 1 materials-16-01070-f001:**
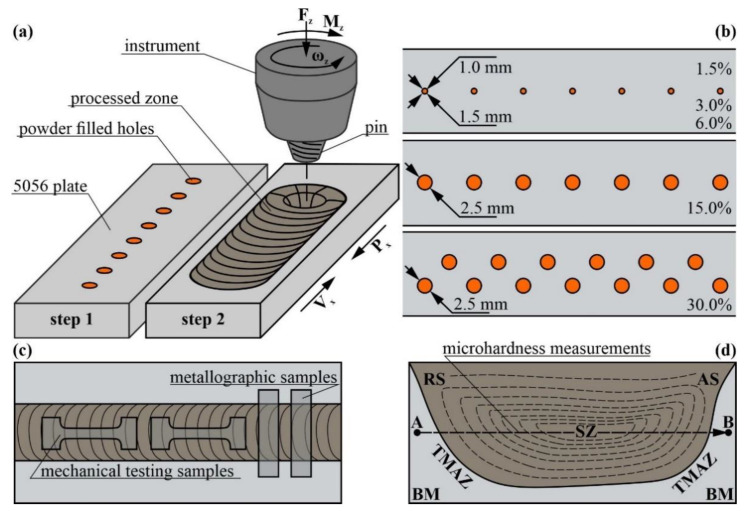
The FSP diagram (**a**), AA5056 plates with drilled holes (**b**), samples for mechanical tests (**c**) and microhardness profile (**d**). RS and AS are the retreating and advancing sides, respectively; SZ and TMAZ are the stir and thermomechanically affected zones, respectively; BM is the base metal. F_z_, M_z_ and ω_z_ are the plunging force, torque and angular speed, respectively. AB is the line along which the microhardness profiles have been obtained.

**Figure 2 materials-16-01070-f002:**
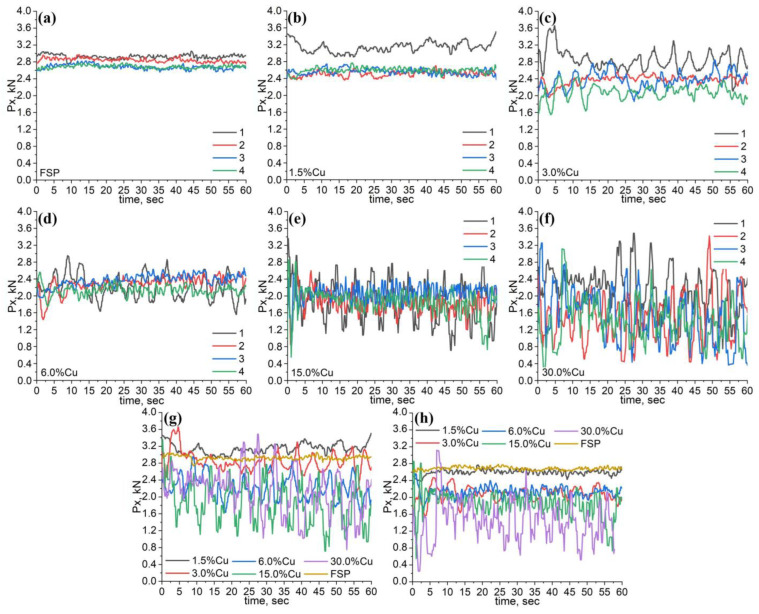
The plasticized metal reaction force P_x_ vs. time for samples with 0 vol.% (**a**), 1.5 vol.% (**b**), 3.0 vol.% (**c**), 6.0 vol.%% (**d**), 15.0 vol.% (**e**), and 30.0 vol.% (**f**) Cu depending on the FSP pass number. The P_x_ dependencies for the first (**g**) and fourth (**h**) passes on all samples.

**Figure 3 materials-16-01070-f003:**
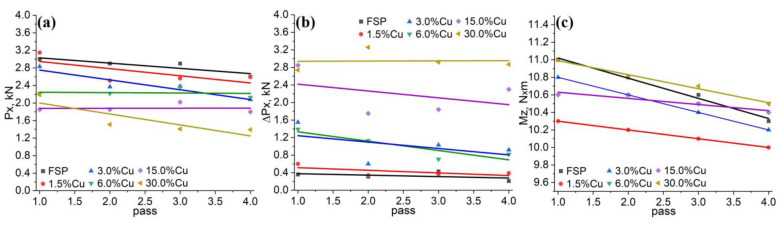
The reaction force (**a**), reaction force scatter (**b**) and torque (**c**) dependencies on the FSP pass number and concentration of copper in samples.

**Figure 4 materials-16-01070-f004:**
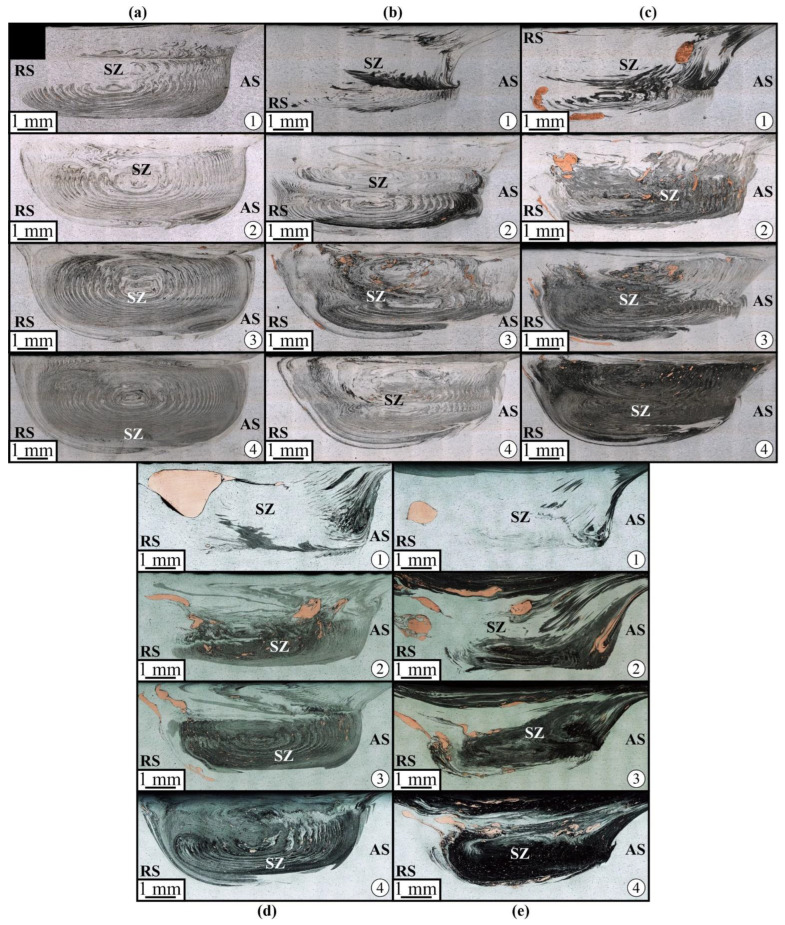
The stir zone macrostructure evolution with the FSP pass number from 1 to 4 views in samples containing 1.5% (**a**), 3% (**b**), 6% (**c**), 15% (**d**) and 30% (**e**) copper.

**Figure 5 materials-16-01070-f005:**
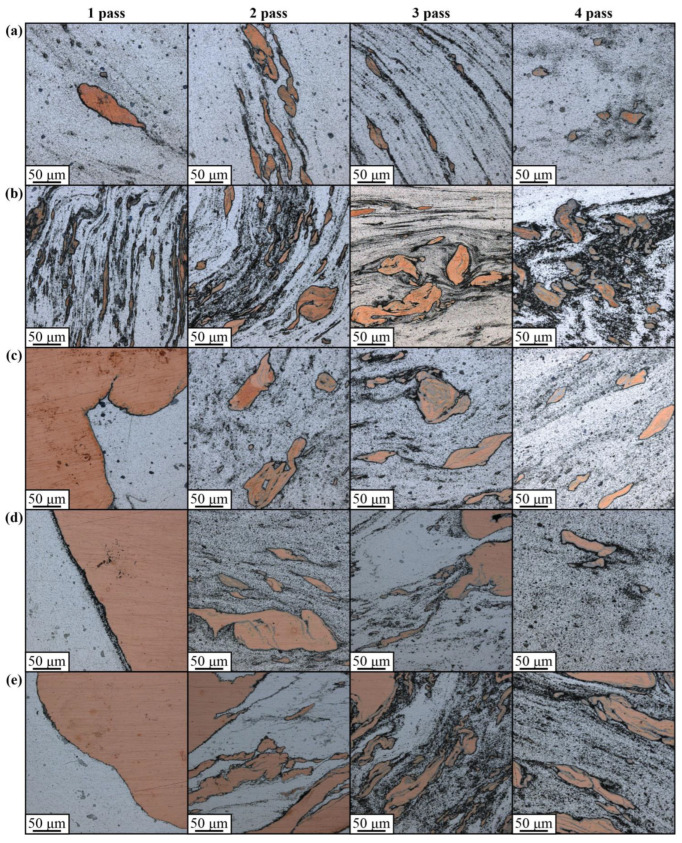
The stir zone microstructure evolution with FSP pass number from 1 to 4 views in samples containing 1.5% (**a**), 3% (**b**), 6% (**c**), 15% (**d**) and 30% (**e**) copper.

**Figure 6 materials-16-01070-f006:**
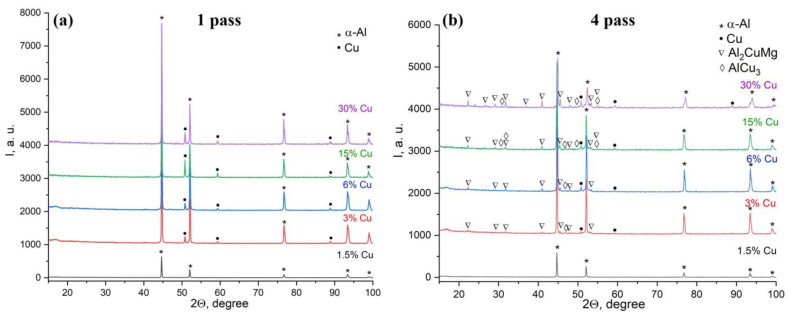
The XRD patterns of composite alloy samples made by 1-pass (**a**) and 4-pass (**b**) FSP.

**Figure 7 materials-16-01070-f007:**
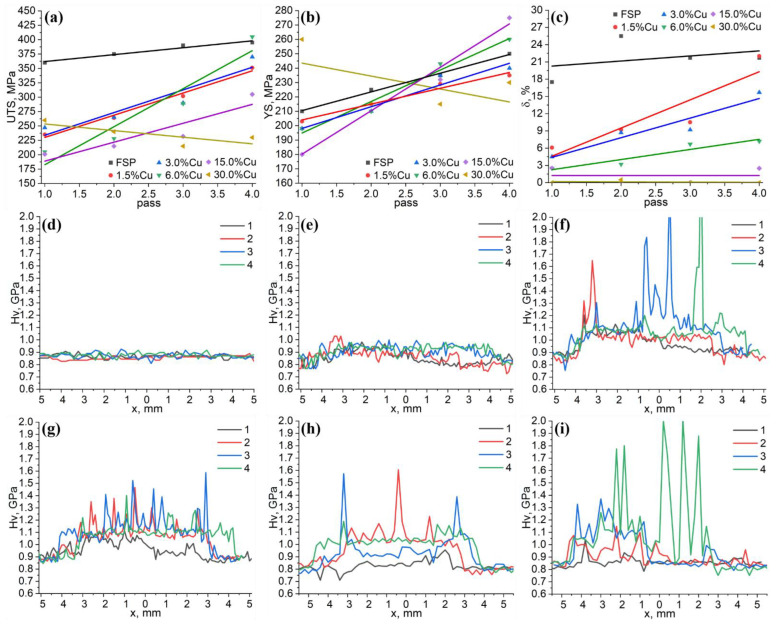
The ultimate tensile stress (UTS) (**a**), yield stress (YS) (**b**), strain-to-fracture (STF) (**c**) dependencies on pass number and copper content, microhardness profiles for AA5056 after FSP (**d**), 1.5%Cu (**e**), 3% (**f**), 6% (**g**), 15% (**h**) and 30% (**i**) copper samples.

**Figure 8 materials-16-01070-f008:**
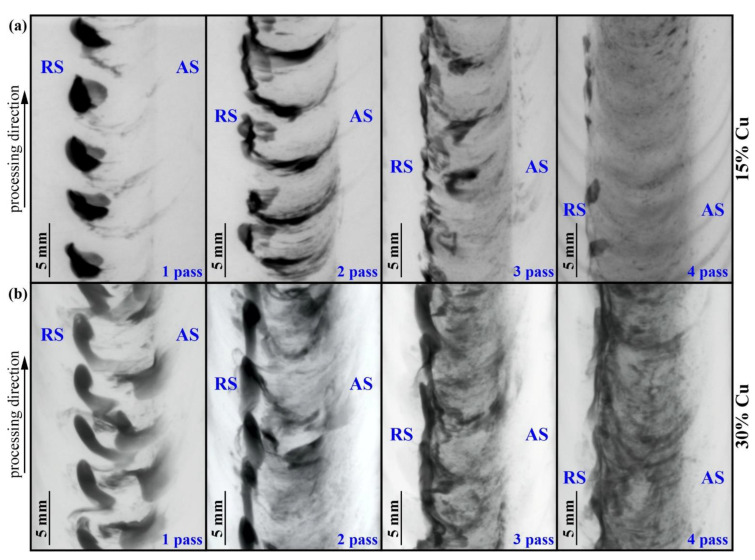
The X-ray computer tomography in-plane images of samples with 15 (**a**) and 30 vol.% (**b**) Cu obtained from samples with 1 to 4 passes after FSP.

**Figure 9 materials-16-01070-f009:**
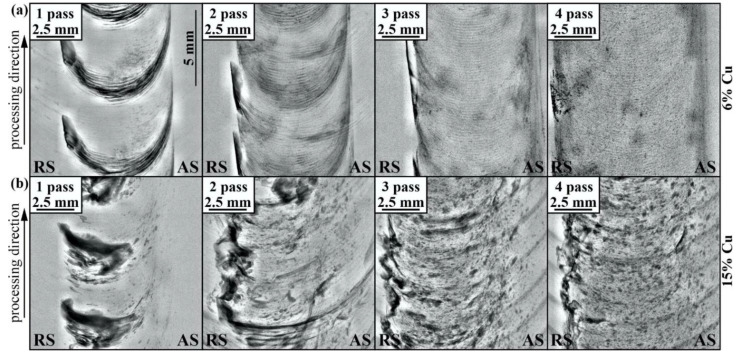
The enlarged X-ray computer tomography in-plane images of samples with 6 (**a**) and 15 vol.% (**b**) Cu obtained from samples with 1 to 4 passes after FSP.

**Figure 10 materials-16-01070-f010:**
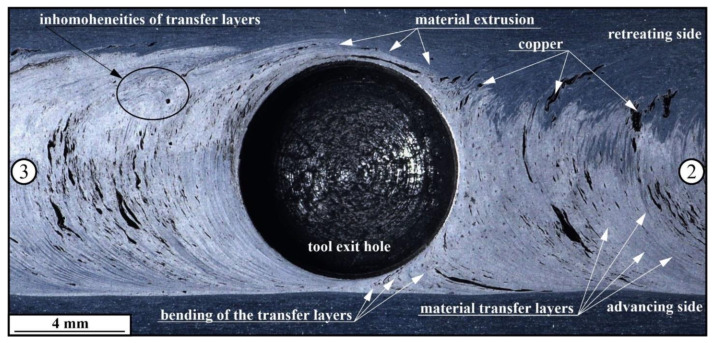
The macrostructure of 6.0 vol.% Cu sample as viewed in a plane parallel to the sample’s horizontal surface and in the vicinity of the tool exit hole left after the 3-pass FSP. Figures ② and ③ stand for marking up the FSP track zones after two and three passes, respectively.

**Figure 11 materials-16-01070-f011:**
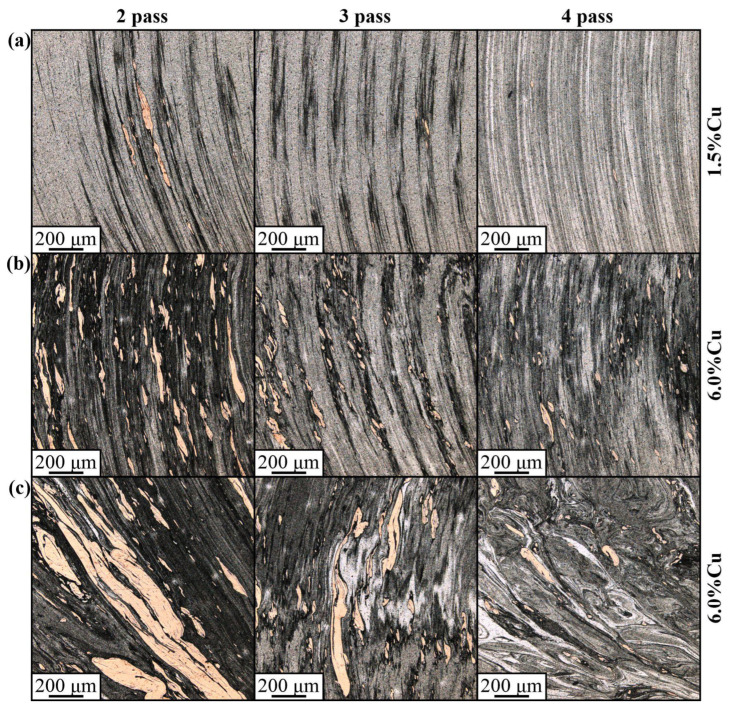
The macrostructures of horizontal plane section in samples containing 1.5 vol.% Cu (**a**) and 6.0 vol.% Cu (**b**,**c**). The macrostructures of intensively stirred zone (**b**) and stagnant zone (**c**).

**Figure 12 materials-16-01070-f012:**
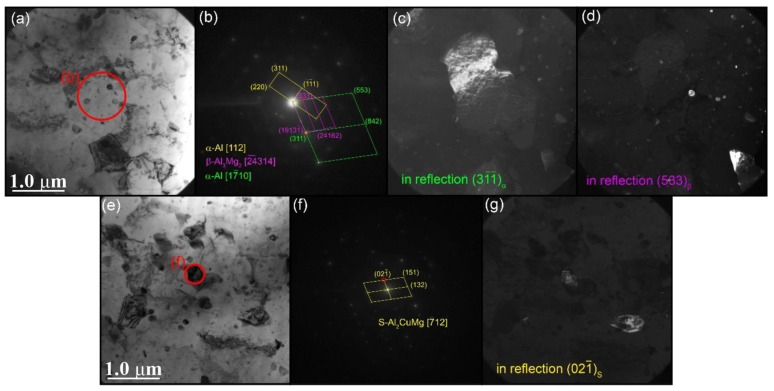
Bright-field (**a**,**e**) with corresponding SAED patterns (**b**,**f**) and dark-field (**c**,**d**,**g**) TEM images of α-Al(Cu) grain (**a**) as well as β-Al_3_Mg_2_ (**d**) and S-Al_2_CuMg (**g**) particles obtained using reflections ($3\bar{1}\bar{1}$), (533) and ($02\bar{1}$), respectively.

**Figure 13 materials-16-01070-f013:**
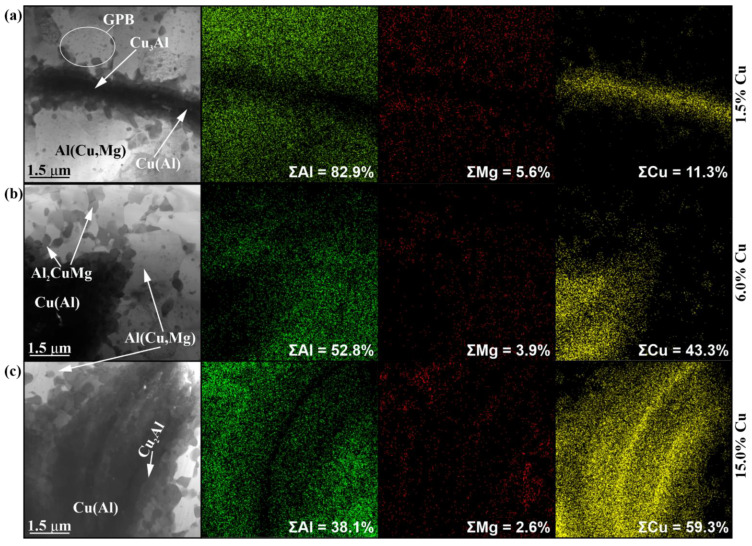
The bright-field TEM images of IMC agglomerate particles in the 4-pass samples after FSP with 1.5 vol.% (**a**), 6.0 vol.% (**b**) and 15 vol.% Cu (**c**) with corresponding EDS maps.

**Figure 14 materials-16-01070-f014:**
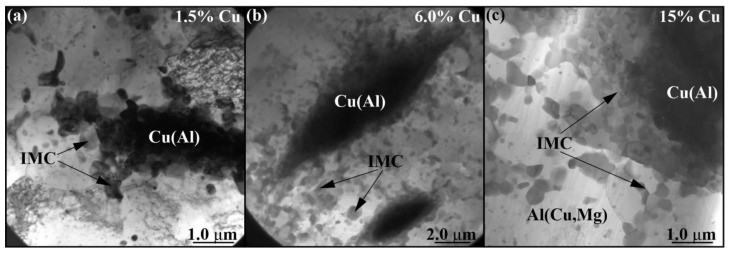
The bright-field TEM images of IMC agglomerate particles in the 4-pass samples after FSP with 1.5 vol.% (**a**), 6.0 vol.% (**b**) and 15 vol.% Cu (**c**).

**Figure 15 materials-16-01070-f015:**
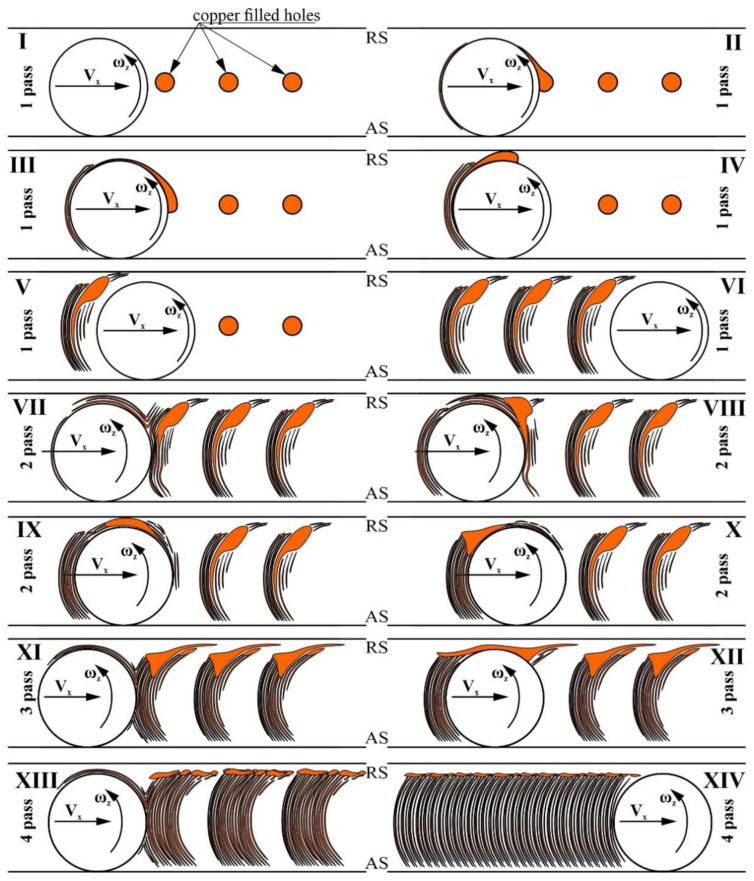
Schematic diagrams showing the FSP stirring and transfer stages that occur during each of the four passes.

**Figure 16 materials-16-01070-f016:**
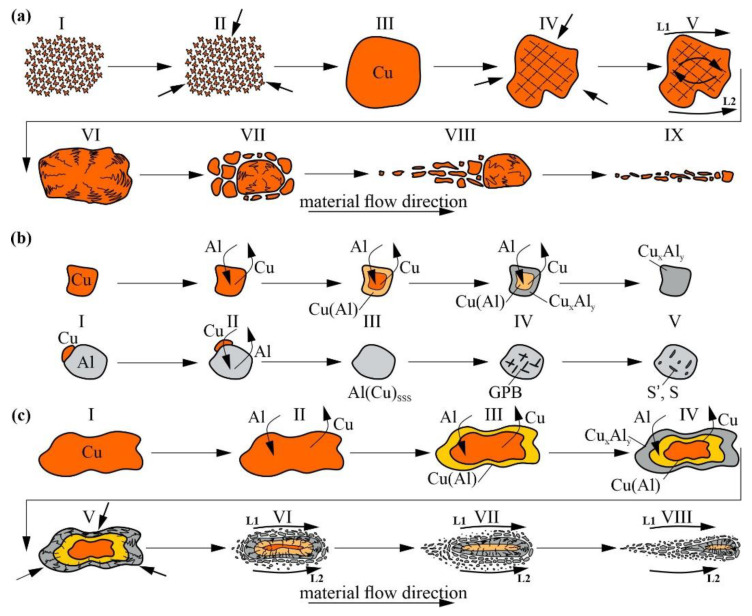
Schematic diagrams showing the effects of thermomechanical compaction (**a**), reaction-diffusion interaction (**b**) and combined mechanical-diffusion interaction (**c**) during FSP on AA5056 + Cu systems.

**Table 1 materials-16-01070-t001:** FSP parameters and mean strength characteristics of composite samples.

Sample	Pass	P_z_, kN	ω_z_, mm/min	V_x_, mm/min	P_x_, kN	M_z_, Nm	UTS, MPa	YS, MPa	STF, %
AA5056	-	-	-	-	-	-	351	160	28.3
AA5056 FSP	1	13.5	500	90	3.00	11	360	210	17.5
	2	13.0	500	90	2.90	10.8	375	225	25.5
	3	12.5	500	90	2.90	10.6	390	235	21.7
	4	12.5	500	90	2.60	10.3	395	250	21.7
AA5056 + Cu 1.5%	1	13.5	500	90	3.15	10.3	235	203	6.1
	2	13.5	500	90	2.51	10.2	264	215	9.2
	3	13.5	500	90	2.56	10.1	302	229	10.5
	4	13.5	500	90	2.59	10	351	235	22.0
AA5056 + Cu 3%	1	13.5	500	90	2.83	10.8	247	198	4.5
	2	13.5	500	90	2.37	10.6	265	210	8.7
	3	13.5	500	90	2.38	10.4	290	235	9.2
	4	13.5	500	90	2.08	10.2	370	240	15.7
AA5056 + Cu 6%	1	13.5	500	90	2.21	10.6	205	198	2.5
	2	13.5	500	90	2.22	10.6	228	210	3.2
	3	13.5	500	90	2.37	10.5	289	243	6.7
	4	13.5	500	90	2.13	10.5	405	260	7.2
AA5056 + Cu 15%	1	12.5	500	90	1.85	10.6	201	180	2.5
	2	13.0	500	90	1.85	10.6	215	215	0.0
	3	13.5	500	90	2.02	10.7	232	232	0.0
	4	13.5	500	90	1.80	10.4	305	275	2.5
AA5056 + Cu 30%	1	11.7	500	90	2.19	11	260	260	0.0
	2	12.5	500	90	1.51	10.8	240	215	0.5
	3	13.0	500	90	1.41	10.7	215	215	0.0
	4	13.5	500	90	1.39	10.5	230	230	0.0

## Data Availability

Data sharing is not applicable to this article.

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
