# Peer review of "Macro- and Microstructure of In Situ Composites Prepared by Friction Stir Processing of AA5056 Admixed with Copper Powders"

_materials, 2023, doi:10.3390/ma16031070_

Round 1

Reviewer 1 Report

Reviewer Comment
Manuscript number: Materials- 2164307

Dear Editor,

The authors have tried to investigate the effect of adding Cu into Al alloys using several (4) passes of Friction Stir Processing (FSP). 

These remarks can be concluded from the authors' submitted manuscript:

1- Acronyms should be identified from the beginning and they should be used throughout the manuscript.

2-Author contributions need clear explanation. How any author significantly and effectively contribute in the processing, methodology, software utlising, writing drafts and supervision of the manuscript ?!

3-Minor English correction and paying attention to a correct punctuation are necessary. Just for example, in the abstract section:

 Microstructural investigations showed that depending upon the concentration of copper and number of FSP passes; different phase compositions have been obtained in the AA5056 stir zone such as α-Al(Cu), α-Cu(Al) solid solutions, Cu3Al and CuAl IMCs , S and S’ Al2CuMg precipitates.

4-Just another example of sentence arrangement and English correction: line 60:

At the same time the drilled hole technique is the simplest and, therefore,.....

Meantime, the drilled hole technique is the simplest (what?). Therefore, it is widely used......

Line 70-72

5-Authors in the line 92-100 gave reliable reasons for selection Al-Cu alloys for testing. Nevertheless, is there any real application in the industry ? How can the authors' selection contribute in significant improvement in mechanical properties for a specific application(s) in the industry?

6-in the Materials and Methods section, the authors states:
powder contents were 1.5, 3.0, 6.0, 15 and 30 vol.% depending...

Can the authors explain why they choose wide range of irrelevant concentrations, especially between 6.0 vol.% to 15 vol.% and between the former to 30 vol.%?

It is better to choose the narrower range, or select other concentrations in between, to verify the effect of Cu addition into Al alloy.

7-Figure 7 d, e, f, g, h,i microhardness unit in HV not MPa.

8-in the related discussion to microhardness, the unit is HV not Hv.

9-how the authors explain the reasons behind hardness fluctuations for figure 7 f and figure 7 i?

10- in the conclusion, authors claim brittle fracture for Al-Cu interfaces. How they can prove that? what kind of brittle fracture mechanism was responsible on this claimed brittleness?

11-references 10,16 and 18 should be corrected.

Author Response

Dear mr./ms. reviewer 
Thak you very much for your comments

Below are the responses to your comments

Best regards, Chumaevskii Andrey

1- Acronyms should be identified from the beginning and they should be used throughout the manuscript.

A: Thank you. Corrected

2-Author contributions need clear explanation. How any author significantly and effectively contribute in the processing, methodology, software utilizing, writing drafts and supervision of the manuscript ?!

A: Please, see the Author Contributions section.

3-Minor English correction and paying attention to a correct punctuation are necessary. Just for example, in the abstract section:

 Microstructural investigations showed that depending upon the concentration of copper and number of FSP passes; different phase compositions have been obtained in the AA5056 stir zone such as α-Al(Cu), α-Cu(Al) solid solutions, Cu3Al and CuAl IMCs , S and S’ Al2CuMg precipitates.

A: Thank you. Corrected

4-Just another example of sentence arrangement and English correction: line 60:
At the same time the drilled hole technique is the simplest and, therefore,.....

Meantime, the drilled hole technique is the simplest (what?). Therefore, it is widely used......

Line 70-72

A: Thank you. Corrected

5-Authors in the line 92-100 gave reliable reasons for selection Al-Cu alloys for testing. Nevertheless, is there any real application in the industry ? How can the authors' selection contribute in significant improvement in mechanical properties for a specific application(s) in the industry?

A: Thank you. Missing information added

6-in the Materials and Methods section, the authors states:
powder contents were 1.5, 3.0, 6.0, 15 and 30 vol.% depending...

Can the authors explain why they choose wide range of irrelevant concentrations, especially between 6.0 vol.% to 15 vol.% and between the former to 30 vol.%?

It is better to choose the narrower range, or select other concentrations in between, to verify the effect of Cu addition into Al alloy.

A: Agree. One of the reasons behind choosing such a wide concentration interval was to study the efficiency of stirring and obtaining homogeneous distribution of copper in the stir zone. Therefore, that high concentrations were used. 

7-Figure 7 d, e, f, g, h,i microhardness unit in HV not MPa.
A: Thank you. Corrected.

8-in the related discussion to microhardness, the unit is HV not Hv.

A: Thank you. Hv stands for  “microhardness”, while the microhardness numbers were measured as  GPa.  

9-how the authors explain the reasons behind hardness fluctuations for figure 7 f and figure 7 i?

A: These high peaks resulted from indenting the hard IMC regions

10- in the conclusion, authors claim brittle fracture for Al-Cu interfaces. How they can prove that? what kind of brittle fracture mechanism was responsible on this claimed brittleness?

A; It is well–known fact that Al-Cu IMCs are brittle and therefore have negative effect on the Al-Cu weld strength. Many publications were devoted to reducing the IMC layer or particle size in the welds by means of, for example, ultrasonic waves. The majority of IMCs are brittle especially if in the form of large particles. The brittle fracture mechanism originates directly from the IMC structure and physical characteristics.    

11-references 10,16 and 18 should be corrected.

A:Thank you. Corrected

Reviewer 2 Report

REVIEWER’S COMMENTS

TITLE OF PAPER:

Macro- and microstructure of in-situ composites prepared by friction stir processing of AA5056 admixed with copper powders.

 REVIEWER’S COMMENTS :

The paper deals with the use of friction stir processing for producing in-situ copper reinforced AA5056 matrix composites. The paper is interesting and it has a novelty. In general, the paper is well written, the methodology used is very good and the data are comprehensively discussed. In my opinion, the paper is accepted with minor corrections as follows.

1.   The percentages of copper powders added to the AA5056 matrix are in the range of 1.5 to 30 vol.%. Please, explain the reasons why the authors choose such compositions. How is the average diameter of copper powders? The authors should provide this information in the paper.

2.   Based on Table 1, the UTS and YS for FSW with no copper additions (AA5056 FSP) are the highest. However, further additions of copper to this AA5056 seem to degrade the strengths. Please, give explanations.

3.   Can intermetallic compounds such as Al2CuMg and AlCu3 form under FSP? Please, add explanations.

4.   The authors should add some references in “Results and Discussion Sections” to support their claims.

5.   Is in-situ composite competitive in terms of mechanical properties in comparison with the metal matrix composite produced using conventional powder metallurgy?

Author Response

Dear mr./ms. reviewer 
Thak you very much for your comments

Below are the responses to your comments.

Best regards, Chumaevskii Andrey

  1. The percentages of copper powders added to the AA5056 matrix are in the range of 1.5 to 30 vol.%. Please, explain the reasons why the authors choose such compositions. How is the average diameter of copper powders? The authors should provide this information in the paper.

A:  One of the reasons behind choosing such a wide concentration interval was to study the efficiency of stirring and obtaining homogeneous distribution of copper in the stir zone. Therefore, that wide concentration range was used. 

  1. Based on Table 1, the UTS and YS for FSW with no copper additions (AA5056 FSP) are the highest. However, further additions of copper to this AA5056 seem to degrade the strengths. Please, give explanations.

A: Table 1 shows that the UTS values obtained on the 4-pass FSP AA5056+1.5% Cu, AA5056+3% Cu and AA5056+6%Cu are 351, 370 and 405 MPa, respectively, i.e higher than 351 MPa of the as-received AA5056 but a bit lower than– 395 MPa of the 4-pass AA5056 However, the 4-pass FSPed AA5056+Cu samples still have some structural and compositional inhomogeneity as well as microscopic defects. For even higher concentrations of Cu there is UTS value degradation related to insufficient intermixing and structural inhomogeneity of that high copper concentrations. It follows from the Table 1 data that the more Cu concentration, the more passes are needed to achieve the acceptable structural and compositional homogeneity. Despite acceptable strength characteristics are achieved after 4-pass FSP on 1.5, 3 and 6vol. % of copper/AA5056 samples, their characteristics can be improved using more FSP passes. The same may be true for the samples with 15 and 30 vol.% Cu.  

  1. Can intermetallic compounds such as Al2CuMg and AlCu3 form under FSP? Please, add explanations.

A: It is not new that various intermetallic compounds may from during FSP. In fact this is a principle that lies underneath  preparing hybrid and in-situ composites  reinforced by particles formed in the stir zone from both added and existing components [https://dx.doi.org/10.3390/met10060772]. Even so-called insoluble iron-containing particles that form in the aluminum alloys after solidification can be strain-dissolved and then precipitated again in FSP [https://dx.doi.org/10.1007/s40194-017-0447-8].  

 Preliminary experiments with FSP intermixing aluminum alloy with copper have shown that even intermetallic eutectics was formed by liquid phase reaction-diffusion [http://dx.doi.org/10.3390/met10060818].

  1. The authors should add some references in “Results and Discussion Sections” to support their claims.

A: Thank you. We added more references and text to the Discussion section. 

  1. Is in-situ composite competitive in terms of mechanical properties in comparison with the metal matrix composite produced using conventional powder metallurgy?

A: The advantage of the in-situ FSP composites is that they may be used for preparing composite coatings directly on any low-melting alloy and using any type of particles or master alloy. Another advantage is the lack of fusion and therefore, the material grains are not melted and strain-induced fine-grained structure may retain.  

It is hardly possible of course to fabricate a composite with the refractory alloy matrix using this FSP method because there is no FSP tool material capable of sustaining high temperatures and severe deformation, adhesion and interaction with the plasticized metal. 

Reviewer 3 Report

Review comments:

This paper is devoted to using a multi-pass friction stir processing (FSP) for admixing 1.5 to 30vol.% copper powders into an AA5056 matrix for in-situ fabricating a composite alloy reinforced by Al-Cu intermetallic compounds (IMC). Macrostructurally inhomogeneous stir zones have been obtained after the first FSP passes, whose homogeneity was improved with the following FSP passes. The stirring of plasticized AA5056 resulted in compacting of the initial copper particle ag- glomerates into large copper particles with simultaneous saturation of them by aluminum. Micro- structural investigations showed that depending upon the concentration of copper and number of FSP passes different phase compositions have been obtained in the AA5056 stir zone such as α- Al(Cu), α-Cu(Al) solid solutions, Cu3Al and CuAl IMCs , S and S’Al2CuMg precipitates. The amount of IMCs increased with the number of FSP passes that resulted in enhanced by 50-55%  microhardness. The effect of the multipass FSP on tensile strength, yield stress and strain-to-fracture was analyzed.

The research is conducted with plentiful microstructural characterization and interesting. The research content is worthy of discussion. There are several points which need to be further clarified by the authors.

1.The English usage and grammar in this manuscript were checked, there are minor grammatical errors throughout the whole manuscript.

2. In the introduction, some literatures are simply listed. A brief review with logical depth is required to emphasis the research significance of this work. The research significance of this manuscript is need re-considered not only by the view of fundamental research but also by the perspective of engineering application.

3. Into 2. Materials and methods

Kindly provide the sample preparation method for microstructure analysis, XRD and TEM, respectively.

4. Into Discussion

It is well known that mechanical properties of the metals are controlled by microstructure factors, such as macro-morphology, micro-morphology, e.g. grain size, phase composition, as well as deformation behaviour, such as dislocation slip, twinning, and deformation bands.

Thus, an in depth study is required for clarifying the relationship between the volume fraction of copper on microstructure and the influence of microstructure factors on the mechanical properties of friction stir processing of AA5056 admixed with copper powders. Tensile tests and hardness test are recommended.

5.Into Conclusions

The conclusion part should be concise and concrete.

Three or four paragraphs describing the main findings of this study should be given before the bullet points of conclusions.

The conclusion part should focus on effect of microstructure factors (grain morphology, IMCs,) on the mechanical properties of friction stir processing of AA5056 admixed with copper powders.

Author Response

Dear mr./ms. reviewer 
Thak you very much for your comments

Below are the responses to your comments.

Best regards, Chumaevskii Andrey

1.The English usage and grammar in this manuscript were checked, there are minor grammatical errors throughout the whole manuscript.

A: Thank you. Corrected.

  1. In the introduction, some literatures are simply listed. A brief review with logical depth is required to emphasis the research significance of this work. The research significance of this manuscript is need re-considered not only by the view of fundamental research but also by the perspective of engineering application.

A: Thank you. Corrected

  1. Into “2. Materials and methods”

Kindly provide the sample preparation method for microstructure analysis, XRD and TEM, respectively.

A: The samples for mechanical and metallographic optical as well as scanning electron mi-croscopic investigations were cut off the workpieces using an EDM machine DK7750 (Suzhou Simos CNC Technology Co., Ltd. Suzhou, China) as shown in Figure 1c. Thin foils for TEM studies have been prepared using an ЕМ-09100IS (JEOL Ltd., Japan) sample preparation system.

  1. Into Discussion

It is well known that mechanical properties of the metals are controlled by microstructure factors, such as macro-morphology, micro-morphology, e.g. grain size, phase composition, as well as deformation behaviour, such as dislocation slip, twinning, and deformation bands.

Thus, an in depth study is required for clarifying the relationship between the volume fraction of copper on microstructure and the influence of microstructure factors on the mechanical properties of friction stir processing of AA5056 admixed with copper powders. Tensile tests and hardness test are recommended.

A: Agree. All the above suggested investigations are necessary despite deformation mechanisms of the AA5056 are well-known. Tensile and microhardess tests have been carried out and the results are shown in Figure 7a-c. However, extra detailed tests will be carried out after determining the optimal concentration of copper and number of passes needed to obtain homogeneous structure and studying the effect of microstructural characteristics.

5.Into Conclusions

The conclusion part should be concise and concrete.

Three or four paragraphs describing the main findings of this study should be given before the bullet points of conclusions.

A: The missing information has been added in the beginning of the Conclusion section.

The conclusion part should focus on effect of microstructure factors (grain morphology, IMCs,) on the mechanical properties of friction stir processing of AA5056 admixed with copper powders.

A: The objective of this work was to determine with the optimal copper concentration and number of passes needed for acceptable homogenization of the stir zone composite. The grain structure of the AA5056 stir zone have been described in a lot of publications as micron-sized recrystallized grains capable to only grain boundary hardening mechanism. The distribution of the IMCs and their role in hardening is more interesting subject that will be covered in further work.

Reviewer 4 Report

The authors in the work  assessed the macro- and microstructure of in-situ composites prepared by friction stir processing of AA5056 admixed with copper powders for admixing 1.5  to 30 vol.%. For research was used of plates were cut off a 5 mm-thick AA5056 rolled sheet and then 3 mm in depth holes were drilled. The holes were filled with the densified copper powder so that the total powder contents were 1.5, 3.0, 6.0, 15 and 30 vol.% depending on the number and diameter of the holes drilled in each of the plates. The authors of the work used research conducted using: optical metallographic microscope, confocal microscope, an SEM instrument, TEM instrument, an X-ray computer tomography, an XRD instrument, test machine and crohardness tester. The results demonstrate  that the mechanical stirring of the plasticized metal and interaction between the FSP tool and copper powders allows obtaining compacted copper metal formations. Also, the acceptable homogeneity of the stir zone was achieved after 4 FSP passes for all  samples except for sample with 30 vol.% Cu. It was demonstrated that the force parameters  correlated with structural evolution of samples and sample containing 6 vol.% Cu and FSPed by 4 passes showed the ultimate tensile stress.

The reviewed article consists of five parts. In the article presented for review, the authors described their research very meticulously and with great care. All the results contained in it were presented in a very clear and legible way for the recipient. The conclusions and the study summary are consistent and well-formulated. However, there are a few details missing that need to be completed:

1.  What kind of friction welding machine used? What was the shape of the stem? e.t.c.

2.  How method a hardness was tested?

3. What was the grain size of the copper powder?

Author Response

Dear mr./ms. reviewer 
Thak you very much for your comments

Below are the responses to your comments.

Best regards, Chumaevskii Andrey

  1. What kind of friction welding machine used? What was the shape of the stem? e.t.c.

A: We used a laboratory friction stir machine designed and built at AO Sespel (Cheboksary) as shown in Fig.1. Also you may see https://doi.org/10.1007/s00170-019-03631-3/

 Pin cone top and bottom  diameters are 6 and 8 mm, respectively. Pin height – 3 mm. Shoulder diameter 16 mm.

  1. How method a hardness was tested?

A: Microhardness profiles were obtained using an Affri DM8 microhardness tester (Affri, It-aly) along the line AB (Figure 1d). The testing of the method was carried out using the corresponding Vickers microhardness test blocks. 

  1. What was the grain size of the copper powder?

A: the mean size of the 99.5 % pure copper powder was 10.5±0.5 μm

Round 2

Reviewer 3 Report

Accept in the present form.